# Verification of the Adequacy of the Portuguese Sustainability Assessment Tool of High School Buildings, SAHSB^PT, to the Francisco de Holanda High School, Guimarães

**Tatiana Santos Saraiva** [1,*]**, Manuela Almeida** [2] **, Luís Bragança** [2] **and Maria Teresa Barbosa** [3]

[1]   International Doctoral Program in Sustainable Built Environment, School of Engineering, Minho University, 4800058 Guimarães, Portugal
[2]   Department of Civil Engineering, School of Engineering, Minho University, 4800058 Guimarães, Portugal
[3]   Department of Civil Engineering, Federal University of Juiz de Fora, 36036 Juiz de Fora, Brazil
*   Correspondence: id4959@alunos.uminho.pt; Tel.: +55-96-98106-1627

**Abstract:** Sustainable development can be achieved through several activities. The building and construction sector (B & C sector) is one of the major industries, and it can play a crucial role in the improvement of the most relevant environmental impacts. Nowadays, there are major concerns related to sustainability in construction. All types of buildings have different technical aspects; therefore, it is required to develop specific sustainability assessment tools. A school building has peculiarities connected to aspects of sustainability, as it is a building planned to offer adequate environments for the education of adolescents and children. This article shows the application of the SBTool methodology developed specifically for Portuguese high schools, SAHSB^PT (Sustainable Assessment for High School Buildings) methodology, that is being elaborated by the first author in her PhD Thesis. This methodology allows architects, engineers and designers to improve sustainability in school buildings, in projects or in the rehabilitation of buildings. The objective of this research is to apply that evaluation tool in order to verify the efficiency of this methodology, as well as to recognize the level of sustainability of the Francisco de Holanda High School Building, in Guimarães, Portugal. The values found in the application of the Sustainable Assessment for High School Buildings in that high schools demonstrate a good result, as the overall value is A, 75% of the total result.

**Keywords:** sustainability assessment tools; high school buildings; Portugal

## 1. Introduction

The concern about environmental impacts in construction has been increasing. This attention should be extended to social, economic, technical and functional issues, always trying to maintain a balance, so the sustainability can be achieved [1]. In order to achieve sustainable development, there must be major changes in society. Academic research has developed and promoted relevant studies related to the transition of sustainability [2].

Some events have driven sustainable development, such as the Paris Agreement at the 21st Conference of the Parties (COP), and Agenda 2030, which brought together 195 countries in search of poverty reduction and protection of natural resources, seeking various solutions to reduce consumption of water and energy, and minimize environmental impacts [3].

The choice of materials and processes used in the construction of a building interferes with the economic and environmental impacts generated by that building. Nowadays, designers have

great environmental awareness, but there is still no systematic use of the sustainable approach in the construction design process [4].

Numerous rating systems for assessing the environmental impact of buildings have been elaborated to support the development of sustainable buildings. Different research institutions proposed tools according to specific needs [5]. The first comparison among major environment assessment tools was published by Crawley and Aho in 1999 [6]. Recently, other important works have been provided from different perspectives about this subject, such as Todd at al. [7], Berardi [8,9] and Abdala et al. [10].

Among these tools, there are some that have been specifically developed for educational buildings, such as Building Research Establishment Environmental Assessment Method for Education [11], Sustainable Building Tool for Kindergarten for 12th grade Schools, SBTool for K-12 Schools [12] and also, Leadership in Energy and Environmental Design Building Design & Construction [13]. These methodologies aim to promote greater sustainability in the school building [14].

There are several sustainability assessment tools in Portugal, such as LEED (Leadership in Energy and Environmental Design), BREEAM (Building Research Establishment Environmental Assessment Method), and other methodologies specific for the country, such as LiderA (Liderar pelo Ambiente para a construção sustentável, Leading for the Environment for Sustainable Construction), NATURA DOMUS and SBTool$^{PT}$ (Sustainable Building Tool, Portugal). However, there is still no methodology for school buildings according to Portuguese reality; therefore, there is a necessity to elaborate a methodology to meet this need.

The methodology used in this research, SAHSB$^{PT}$ methodology, is an evaluation tool that is being adapted by the author in her PhD thesis, and is described in the article "Adaptation of the SBTool for Sustainability Assessment of High School Buildings in Portugal—SAHSB$^{PT}$" [15]. That methodology was developed having the SBTool$^{PT}$ as the basis for assessment, since the SBTool$^{PT}$ has already been adapted to other building typologies, according to the Portuguese reality, such as SBTool$^{PT}$-H (Homes) [16], SBTool$^{PT}$—Urban (Urban Planning) [17] and SBTool$^{PT}$-STP for Office [18].

Portugal has a population of 10,320 million and an area of 92,212 km$^2$. It has 1819 schools, with 616 high schools, and 1,629,116 students, from which 307,984 are in high schools [19]. This country incorporates a total of 477 schools whose construction started at the end of the XIX century. Of all the schools across the country, about 23% were built up until the end of the 1960s, and the remaining 77% corresponds to the period of expansion of the school network. Thus, 46% of the schools was built in the 1980s, due to the increase of the compulsory schooling, from six to nine years [19].

On 3 January 2007, the Resolution of the Council of Ministers number 41/2007 approved the Program of Modernization (Programa de Modernização de Escolas de Educação Secundária, PMEES). This program intended to improve the Portuguese educational facilities that were underdeveloped when compared to the European standards, and to provide a gratifying and motivating learning environment to children and youth [20]. This Decree-Law deals with the planning, development, management, and implementation of the modernization program of the public high schools. It has three main objectives, to [20]:

- Improve buildings, encouraging a culture of learning;
- Open the school to the community, redirecting the school towards urban environments;
- Create an efficient and effective system of management of buildings.

The Program for the Modernization of Schools for Secondary Education (Programa de Modernização das Escolas com Ensino Secundário, PMEES) aimed to requalify the buildings of Portuguese high schools. The PMEES included interventions in 74% of all high schools, 332 of the 447 high schools in Portugal [20]. About 170 of the high schools reformed by the Empresa Parque Escolar (EPE) will be delivered by the end of 2019. That is around 38% of the Portuguese high schools and 51% of the high schools planned to be reformed by *Parque Escolar* [21].

## 2. Materials and Methods

The object of research of this paper, Francisco de Holanda High School (FHHS), was built in 1894 to be an industrial school. In 1959, a renovation took place, with the creation of its main building, and from 2011 to 2013, EPE made a major renovation in this school. FHHS is an example of high school building standard in Portugal [22]. The high school students age varies between 15 and 18 years old (10th to 12th grade), corresponding to the third cycle of basic education. In Portugal, the classes in high schools usually start at 9 in the morning and end at 3.30 in the afternoon, with breaks for lunch and in the middle of the morning. Some schools work by turns: morning, from 8 a.m. to 1 p.m., or afternoon, from 1.15 p.m. to 6.15 p.m., 25 h of lessons per week [23].

This work sought to adapt the Methodology SBTool in Portuguese high school buildings, SAHSB$^{PT}$ methodology, supporting architects and engineers to increase sustainability in these types of buildings, at the time of project design or in the rehabilitation of constructions. This is significant because there is no specific methodology elaborated in Portugal for this purpose.

The methodology SBTool$^{PT}$ STP for Office Buildings was selected as the basis for this methodology since it allows a better possibility of adjusting the indicators evaluated in each dimension, considering location and the typology of the building. This methodology shows significant flexibility and decreases the subjectivity, which makes it more suitable when compared to other extensive and rigid methodologies [18]. The methodologies, SAHSB$^{PT}$ methodology and SBTool$^{PT}$ STP for Office Buildings, are for service buildings. Furthermore, as the SBTool$^{PT}$ was developed in Portugal, it is the best suited to the Portuguese reality.

SAHSB$^{PT}$ methodology can bring great benefits to school buildings, namely reducing costs and consumption, increasing the awareness of the sustainable user, assisting in the decision-making of architects and engineers during all phases of construction and maintenance of the building, promoting sustainable practices, and thereby reducing the environmental and economic impact of school buildings [24].

The application of this tool in the FHHS will verify if the indicators of the SAHSB$^{PT}$ methodology are adequate. Also, with the results of this application, it can be realized that the situation of the high schools in Portugal regarding the level of sustainability as FHHS is representative of this kind of buildings.

*Structure of the SAHSB$^{PT}$ Methodology*

The SAHSB$^{PT}$ methodology has the major objective of decreasing mistakes in the evaluation process, allowing the evaluator to quantify the performance of the high school building (Sustainability Level—SL). Sustainable indicators support qualitative and quantitative assessments of issues related to the economic, environmental and social development of a building, combining approaches guided by national and international standards, with specialists from different areas [25].

The SAHSB$^{PT}$ methodology, published in a previous article by the authors [15], has three dimensions, 11 categories and 23 indicators which are listed in the following sentences.

Environmental Dimension:

- Category 1 (C1)—Climate Change and Outdoor Air Quality: Life cycle Environmental Impacts (I1) and Heat Island Effects (I2);
- Category 2 (C2)—Biodiversity and Land Use: Land Use Efficiency (I3) and Product With Organic Basis Certificate (I4);
- Category 3 (C3)—Energy: Energy Consumption (I5), Renewable Energy (I6) and Commissioning (I7);
- Category 4 (C4)—Materials, Residues and Resources Management: Reuse and Recycle of Materials (I8), Environmental Management Plan (I9) and Flexibility and Adaptability (I10);
- Category 5 (C5)—Water: Water Consumption (I11), Water Treatment and Recycling (I12) and Collection and Reuse of Rainwater (I13).

Social Dimension:

- Category 6 (C6)—Comfort and Health of Users: Indoor Air Quality (I14), Thermal Comfort (I15), Visual Comfort (I16), Acoustic Comfort (I17) and Ergonomic Comfort (I18);
- Category 7 (C7)—Accessibility: Mobility Plan (I19);
- Category 8 (C8)—Occupants Security: Occupant Security and Safety (I20);
- Category 9 (C9)—Education for Sustainability Awareness: Sustainability Awareness (I21);
- Category 10 (C10)—Accessibility to Public Transport: Accessibility to Public Transport (I22).

　　Economic Dimension:

- Category 11 (C11)—Life Cycle Cost: Life Cycle Costs (I23)

The procedure of the application of the SAHSB$^{PT}$ methodology is the following:

- Phase 1: Quantification of each indicator (23 indicators), includes quantification of the indicators and standardization of indicators.

A value is defined by the standardization that determines the level of sustainability of the building according to the benchmarks of each indicator. The standard practice is the lowest value to consider a high school building sustainable. This value is established at the lower levels prescribed in the requirements and regulations of construction in Portugal. The best practice is the value given by designers with prestige in the field of sustainable building; or the data created by scientific research on the issue of each indicator or through national or international standard [16]. In this process, the equation of Diaz-Baltero [26] is used.

$$\overline{P_I} = \frac{P_I - P_{I*}}{P_I* - P_{I*}} \tag{1}$$

$P_I$ Result of the quantification of each indicator.
$P_I*$ Best practice value.
$P_{I*}$ Standard practice value.
$\overline{P_I}$ Standardization value of each indicator

　　With the purpose of simplifying the comprehension of the results, the standardization values ($\overline{P_I}$) are converted into a qualitative scale. The results range from E (low level of sustainability) and A (high level of sustainability), when level D to the standard practice and level A relates to the best practice, as shown in Table 1.

- Phase 2: Quantification of each category (11 categories): The quantification of the individual performance of each category is the arithmetic average of all indicators related to all categories. In this work, there are 11 categories. The aggregations of the indicators are made according to the SBTool$^{PT}$-H methodology [21]. The "($P_I$)" in Table 1 intends to evaluate the result of each category.
- Phase 3: Quantification of the dimensions of sustainable development (social, environmental and economic): Quantification of Dimensions of Sustainable Development is the arithmetic average of the results obtained in the categories, in order to quantify three dimensions: social, environmental and economic aspects of the area. In order to reach the conclusion on the evaluation for each dimension, Table 1 should be used, as previously presented. The "($P_I$)" in Table 1 aids to evaluate the results for all dimensions. The value of these dimensions are environmental dimension (35%), social dimension (35%) and economic dimension (30%).
- Phase 4: Quantification of the Sustainability Level (NS): The global performance condenses the performance of the dimensions in a single value. The best solution is the one that presents a balance between the various macro indicators. In order to reach the conclusion of the evaluation, Table 1 should be used, as previously presented.
- Phase 5: Sustainability Certificate: the certificate for the building that was submitted to the application of this methodology is demonstrated. The certificate informs the value of all dimensions and the

value of the global performance. The categorization of levels was performed through a six-level scale: from E (low level of sustainability) to A + (high level of sustainability) according to the SBTool$^{PT}$ H methodolog

**Table 1.** General evaluation for dimensions, indicators and categories of Sustainable Assessment for High School Buildings (SAHSB$^{PT}$).

| Level | Conditions | Please Check the Level Reached (✓) |
|---|---|---|
| A$^+$ | $\overline{P_I} > 1.0$ | |
| A (Best Practice) | $0.7 < \overline{P_I} \leq 1.0$ | |
| B | $0.4 < \overline{P_I} \leq 0.7$ | |
| C | $0.1 < \overline{P_I} \leq 0.4$ | |
| D (Standard Practice) | $0.0 \leq \overline{P_I} \leq 0.1$ | |
| E | $\overline{P_I} < 0.0$ | |

The city of Guimarães is located in the north of Portugal, 57 km from the city of Oporto. The climate is temperate and warm, with an average annual relative humidity of 81% and an average annual temperature of 14 °C. The Francisco de Holanda High School was built in 1864 to be an Industrial School. In 2011, it underwent a major renovation, carried out by the Portuguese public company Empresa Parque Escolar (EPE), with the purpose of modernization [20].

The FHHS is a public school, which offers several technical courses in the areas of business, electronics, design, mechatronics, management and programming of computer systems, also partnering with reputable companies, thereby helping students to reach their first job. The school conducts activities open to society, as happens in Semana Aberta (Open Week), promoting lectures and workshops, with varied themes, and courses also open to the community.

The research was conducted with the participation of students through questionnaires related to environmental comfort and environmental awareness, which were applied to these students to verify their level of satisfaction. An interview as well as a site visit were conducted with the collaboration of the administrative director coordinator of the school in question, in which all school environments were visited to check materials, school operation, lighting, ventilation and thermal details, as well as school operation and student safety. The management and control system provided by the Empresa Parque Escolar was also demonstrated, which provides continuous information about the existing problems in the school, being controlled through the school director cell phone, thus contributing to the faster and more efficient solution of lighting and thermal problems, among others.

Specific instruments were also used to measure elements (decibel meter, lux meter, thermometer) related to acoustics, thermal and lighting comfort. For the verification of the technical details, the Empresa Parque Escolar provided several documents concerning electrical, hydraulic and construction materials, and coating processes and materials, as well as floor plans and facades, the declaration of regulatory compliance and the descriptive memorial. The website of the Empresa Parque Escolar (Portal Parque Escolar) was also consulted for further information.

## 3. Results and Discussion

This section presents the results of the application of the methodology in the Francisco de Holanda High School (FHHS), aiming to validate the adequacy of the benchmarks and their applicability in practice to the context of the Portuguese school buildings.

### 3.1. Francisco de Holanda High School, Guimarães, Portugal

The application of the SAHSB$^{PT}$ methodology was carried out in Francisco de Holanda High School, located in Alameda Dr. Alfredo Pimenta, 4814, Azurém, Guimarães [21].

The renovation works of the Francisco de Holanda High School were carried out through the maintenance of the structural and modification conditions. The main block (light color in Figure 1)

was kept in its formal identity, with the pedagogical, administrative and social areas and the library. In this building, the largest facades are at 230 ° north, northwest and southeast facades.

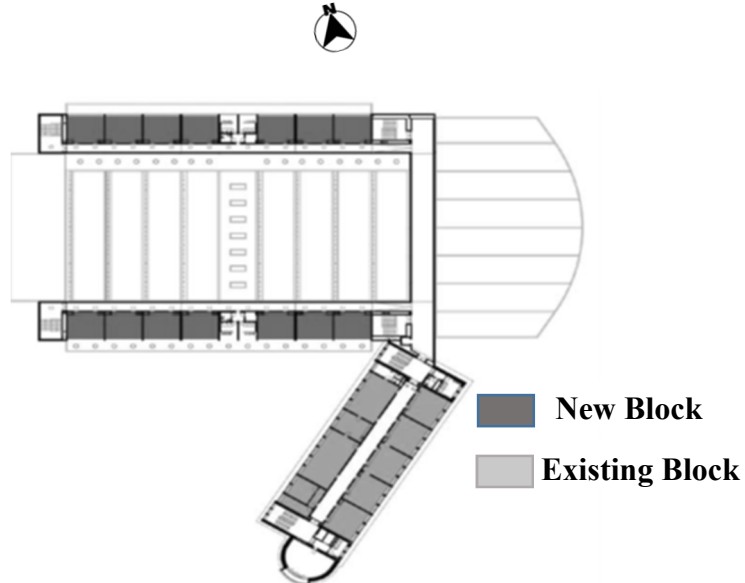

**Figure 1.** Francisco de Holanda High School [21], adapted by the authors.

The building where the new classrooms were located resulted in the addition of two new volumes located on the sides (dark gray in Figure 1), and absorbed the area corresponding to classrooms, science and technology laboratories, and workshops. The largest facades are north and south, as these facades form an angle of 105 ° north. The building materials of the facades, the interior walls, the windows and the doors used in the FHHS are defined in the supplementary material attached to this article.

The new construction has different environments, such as a large cafeteria that has a living room and a multipurpose area with a large triangular patio. In order to reduce the visual impact of the playground coverage, the terrain was lowered and framed by high walls that delimit the school environment. The external area of the school has also been modified to better suit the needs of the students.

The school has a centralized system that controls all energy-related components, such as air conditioning, ventilation and water heating, among others. This control is in the hands of the deputy principal of the school, who receives up-to-date information that facilitates problem solving, and helps to reduce consumption and waste.

A questionnaire regarding the environmental comfort level of Francisco de Holanda High School was applied to students from 15 to 18 years old, from school on 11 January 2017, in winter. The total number of students is 1400, and 19.21% of the students, 269, answered the questionnaires.

The temperature in the classrooms varied between 18 °C and 22 °C during the survey period. Regarding air quality, according to the Regulatory Compliance Statement for this school building, it has a good result. There is a mechanical system, thereby improving the space thermal comfort.

Temperatures are maintained with little variation throughout the year through an air conditioning system in every classroom. Another factor that helps maintain the temperature of the classrooms, and thereby assisting energy efficiency are the window panes. As for the windows, double glazing and laminated glasses are used, separated by an air cavity (12 mm) of dehydrated air.

Wide windows (L = 2 m, H = 1 m) and skylights in darker places benefit from the entry of natural light. Artificial lighting is suitable for the learning environment and is achieved through the use of fluorescent lamps and metal halide lamps.

The material used in the floor of the classrooms is formic as well as the one used for the chairs and tables. The wall paint is bright white, and the windows are double glazed. The use of these materials

results in a bad acoustic performance with a high reverberation time because they are smooth and hard. Consequently, the reverberation time is 0.75 s, a value that is higher than that allowed by the European Comfort Standard EN 15251 (0.60 s). Internal noise ranges from 54 and 71 dB, a value higher than that allowed by the European Comfort Standard EN 15251 (30 to 45 dB).

The questionnaire consisted of six questions about the environmental comfort level of the students within the classroom; the sixth question is about the overall level of comfort. The results of this survey are the following:

61% of students are comfortable regarding temperature;
78% of students are comfortable about the lighting environment;
54% of the students are comfortable concerning the noise level;
72% of students are comfortable regarding air quality;
27% of the students are comfortable with ergonomics;
56% of the students are comfortable concerning to global comfort aspects.

The level of satisfaction of the students regarding thermal, lighting, air quality and thermal comfort was good, but the satisfaction was lower concerning ergonomic and acoustic comfort.

### 3.2. The Result of the Application of the Methodology

The application of the SAHSB$^{PT}$ methodology at Francisco de Holanda High School was done through student questionnaires, forms answered through document analysis, and in situ measurements, described in detail in the author's doctoral thesis. Only indicator 1, the most complex, is in the supplementary material in this article.

The best value found for an indicator or category may be the highest or the lowest value, depending on the analysis that is performed. To avoid the scale effects, the formula of Diaz Balteiro [26], Equation (1), as described in subchapter 2.1 of this article, is used, aiming to normalize the indicators and categories of this methodology. For the application of the methodology, the answers acquired by the guide were always taken into consideration, with the application of that formula [26], considering the best and conventional practices. The results of the indicators, categories and dimensions related to the application of the SAHSB$^{PT}$ methodology in Francisco de Holanda high school are presented below.

Environmental dimension

C1 Category: Climate Change and Outdoor Air Quality

I1. Life Cycle Environmental Impacts

To find the life cycle (LC) environmental impact of the high school building, according to SAHSB$^{PT}$ methodology, it is necessary to identify the construction elements of the building and their measurements; identify the LC environmental impact value of each construction solution used and its maintenance operations. Then, multiply the value of each environmental impact by the amount of different elements and measurements of the building [15,16,18]. The calculation is demonstrated in ANNEX 1. Based on this calculation, the value given to this indicator is 0.75%.

I2. Heat Island Effect

The reflectance resulting from the color and the type of material used in the façade was analyzed, as well as the percentage of green spaces on the ground [15,16,18]. The SAHSB$^{PT}$ methodology uses Solar Reflectance Index (SRI) [27] to evaluate this indicator. The façades of the Francisco de Holanda High School are white or with light colors, and the roofs are made of ceramic material. These materials benefit the mitigation of heat island effects [21].

Calculation of Indicator I2:

$$P_{REF} = \frac{A_{GS} + A_{REF}}{A_{TOT}}, P_{REF} = \frac{4178 + 5557}{12,810} = 0.77 \tag{2}$$

$A_{TOT}$—Total land area in horizontal projection (m$^2$);
$A_{GS}$—Area of green spaces of the building in horizontal projection (m$^2$);

$A_{REF}$—Constructed area in horizontal projection (not covered outdoor decks and roofs) with reflectance equal to or greater than 60%;

$P_{REF}$—Percentage of plan area with a reflectance lower than 60%;

$\overline{P_{REF}}$—Normalized value of the indicator Heat Island Effect.

$$\overline{P_{REF}} = \frac{P_{REF} - P_{REF*}}{P_{REF}^* - P_{REF*}}, \frac{P_{REF} - 30\%}{90\% - 30\%} = \frac{76\% - 30\%}{90\% - 30\%} = 0.77\% \tag{3}$$

Based on this calculation, the value given to this indicator is 0.77%.

Calculation of Category 1

In Table 2, it is calculated the percentage of Category 1, by the sum of the indicators included in this category, and the percentage relative to the total value, 1.

$$\sum \text{Value} [A] \times [B] = \sum \text{Indicator Evaluation} [A] \text{of C1, C1 1, } 5.31 = 7.00 \text{ C1} = 0.76\%, \text{ C1 1} \tag{4}$$

**Table 2.** Calculation of the percentage of Category 1.

| Category | Indicator | Weight of Indicator (%) [A] | Evaluation of Indicator [B] | VALUE (%) [A]×[B] |
|---|---|---|---|---|
| C1 Climate Change | I1 Life Cycle Environmental Impacts | 4.00% | 0.75 A | 3.00% |
| Outdoor Air Quality | I2 Heat Island Effects | 3.00% | 0.77 A | 2.31% |
| Sum | | 7.00% | | 5.31% |

Based on this calculation, the value given to the category is 0.76%.

C2. Category: Biodiversity and Land Use

I3. Land Use Efficiency

Empresa Parque Escolar (EPE) aims to open the school to the community, creating functional spaces to be used for sports, social, cultural and leisure activities. EPE also seeks the flexibility and adaptability of the entire school environment to maximize its use [21]. In this indicator, the net usable area, the gross area, the area of implantation, the area of the plot and the number of occupants of the building are analyzed [15,16,18]. In addition, Francisco de Holanda High School is open to the community just for cultural activities.

$$T_{ERO} = \frac{A_U \times C_O}{A_E \times A_I \times A_P}, T_{ERO} = \frac{13,877 \times 2874}{16,326 \times 7253 \times 12,810} = 0.0000263 \tag{5}$$

$T_{ERO}$—Efficiency Ratio on Territorial Occupation;

$A_E$—Gross external area: It is the total area of the fractions, as the outside perimeter of the exterior walls and axes of the partition walls and accessory includes local private balconies, and the share which corresponds to it in the circulations of the building;

$A_I$—Area of implementation: Area of the vertical projection on the ground of constructions, considering the upper surface of the walls including basements and outbuildings. Exceptions are only influenced by the following elements: balconies, parapets, overhangs and terraces;

$A_P$—Plot area or lot area: Land area in vertical projection, bounded by neighboring land or public roads, resulting from the sum of the deployment area of the building and the patio area;

$A_U$—Net usable area: It is the sum of the compartment areas of school buildings, excluding interior circulation and bathrooms, and it is measured by the inner perimeter of the walls that limit the school building, discounting the interior walls and ducts;

$C_O$—Number of students inside the classroom;

$\overline{T_{ERO}}$—Normalized value of the Indicator Land Use Efficiency.

$$\overline{T_{ERO}} = \frac{T_{ERO} - T_{ERO*}}{T^*_{ERO} - T_{ERO*}}, \frac{0.0000263 - 0.00002 \text{ c}/ m^4}{0.00003 \text{ c}/m^4 - 0.00002 \text{ c}/ m^4} = 0.63\% \quad (6)$$

Based on this calculation, the value given to this indicator is 0.63%.

I4. Product with Organic Basis—Certificate

This indicator is related to the cost of wood or organic materials with environmental certification [15,16,18]. Through the analysis of the materials elaborated by the Empresa Parque Escolar (EPE), it was observed that there is no specific concern on this subject [21]. Therefore, there was no concern regarding organic certificates during the work. Based on this information, the value given to this indicator is the result 0%—E.

Calculation of Category 2

In Table 3, the percentage of Category 2 is calculated, by the sum of the indicators included in this category, and the percentage relative to the total value, 1.

$$\sum \text{Value } [A] \times [B] = \sum \text{Indicator Evaluation } [A] \text{ of C2, C2 1, } 2.52 = 5.00 \text{ C2} = 0.50\%, \text{C2 1} \quad (7)$$

**Table 3.** Calculation of the percentage of Category 2.

| Category | Indicator | Weight of IND. (%) [A] | Evaluation of IND. [B] | VALUE (%) [A]×[B] |
|---|---|---|---|---|
| C2. Biodiversity and Land Use | I3 Land Use Efficiency | 4.00% | 0.63—B | 2.52% |
| | I4 Product with Organic Certificate | 1.00% | 0—E | 0% |
| Sum | | 5.00% | | 2.52% |

Based on this calculation, the value given to this category is 0.50%.

C3. Category: Energy

I5. Energy Consumption

This indicator has to do with the energy consumption values per year associated to electricity (EE) and gas (EG). It is based on the procedures of RECS [28] for energy consumption. Energy efficiency is one of the major objectives of the Empresa Parque Escolar, seeking to "ensure the energy efficiency of buildings in order to reduce operating costs" [21]. According to the Regulatory Compliance Statement [14] of this school building, the total amount spent on energy is 18.61 Kgp/m$^2$-year.

$$\overline{E_C} = \frac{E_C - E_{C*}}{E^*_C - E_{C*}}, \overline{E_C} = \frac{186 - 310}{231 - 310} = 0.80\% \quad (8)$$

$\overline{E_C}$—Normalized value of the Indicator Energy Consumption.

Based on this calculation, the value given to this indicator is 0.80%.

I6. Renewable Energy

This indicator refers to the values of renewable energy consumption per year related to Photovoltaic and Solar Panels, based on the procedures of RECS [28]. In the high school buildings refurbished or built by Empresa Parque Escolar (EPE), there is a concern with renewable energy, through solar collectors for heating water for the kitchen and bathrooms [21]. The total energy produced by this system is 30421 kwh/year, according to the Regulatory Compliance Statement [29].

$$\overline{E_{REN}} = \frac{E_{REN} - E_{REN*}}{E^*_{REN} - E_{REN*}}, \frac{48\% - 30\%}{60\% - 30\%} = 0.60\% \quad (9)$$

$\overline{E_{REN}}$—Normalized value of the Indicator Renewable Energy.

Based on this calculation, the value given to this indicator is 0.60%.

I7. Commissioning

This indicator is performed through a form that assesses the commission team in aspects such as: schedule that defined the main milestones and dates; budget for the installation and acquisition of energy; documentation relating to the energy and building system; plan for the control of mechanical systems and performance verification [15,16,18]. Empresa Parque Escolar offers a suitable administration of all mechanical systems in the high schools, being all of them controlled through a system provided by EPE [21]. These systems are controlled through computers and the personal cell phone of the school administrator. Based on this information, the value given to this indicator is 1.0%.

Calculation of Category 3

In Table 4, the percentage of Category 3 is calculated, by the sum of the indicators included in this category, and the percentage relative to the total value, 1.

$$\sum \text{Value [A]} \times \text{[B]} \ = \ \sum \text{Indicator Evaluation [A] of C3, C3 1, 8.8} = 11.0 \text{ C3} = 0.80\%, \text{ C3 1} \quad (10)$$

**Table 4.** Calculation of the percentage of Category 3.

| Category | Indicator | Weight of IND. (%) [A] | Evaluation of IND. [B] | Value (%) [A]×[B] |
|---|---|---|---|---|
| C3. Energy | I5. Energy Consumption | 5.00% | 0.80—A | 4.00% |
| | I6 Renewable Energy | 3.00% | 0.60—B | 1.80% |
| Sum | I7 Commissioning | 3.00% | 1.10—A* | 3.00% |
| | | 11.00% | | 8.80% |

Based on this calculation, the value given to this category is 0.80%.

C4. Materials, Solid Residues, and Resources Management.

I8. Reuse and Recycle of Materials

This indicator is related to the price value of the elements of the building that are pre-existing in the construction and is reuse, and the materials from deconstructions. [15,16,18]. There was no concern regarding the reuse of materials or products or the material with recycled content used in the construction of the FHHS building. Based on this information, the value given to this indicator is 0.

I9. Environmental Management Plan

This indicator is performed through a form that assesses the environmental management and monitoring system; training of occupants, management of solid waste treatment; and management of consumption of the products in the use phase [15,16,18]. One of the objectives of Empresa Parque Escolar is "to create an efficient and effective system of building management. An Environmental Management System is used" [21].

$$\overline{P_{MS}} = \frac{P_{MS} - P_{MS*}}{P_{MS}^{*} - P_{MS*}}, \frac{130\% - 30\%}{120\% - 30\%} = 1.10\% \quad (11)$$

$\overline{P_{MS}}$—Normalized value of the Indicator Environmental Management Plan.

Based on the calculation, the value given to this indicator is 1.10%.

I10. Flexibility and Adaptability

This indicator is performed through a form that evaluates the air conditioning (location of ducts and size of equipment), ventilation systems, electrical and communications systems (ducts location), water system and plumbing (location), and the modularity of the compartments [15,16,18]. One of the objectives of Empresa Parque Escolar is to "ensure flexibility and adaptability of school and non-school spaces in order to maximize their use and minimize future investments" [21]. Despite the use of the

Environmental Management System, the type of construction process still follows the traditional style, with few concerns regarding flexibility and adaptability.

$$\overline{P_{FA}} = \frac{P_{FA} - P_{FA*}}{P_{FA}^* - P_{FA*}}, \frac{16\% - 11\%}{25\% - 11\%} = 0.34\% \tag{12}$$

$\overline{P_{FA}}$—Normalized value of the Indicator Flexibility and Adaptability.

Based on this calculation, the value given to this indicator is 0.34%.

Calculation of Category 4

In Table 5, the percentage of Category 4 is calculated, by the sum of the indicators included in this category, and the percentage relative to the total value, 1.

$$\sum \text{Value } [A] \times [B] = \sum \text{Indicator Evaluation } [A] \text{of C4, C4 1, } 2.71 = 5.00 \text{ C4} = 0.54\%, \text{ C4 1} \tag{13}$$

**Table 5.** Calculation of the percentage of Category 4.

| Category | Indicator | Weight of IND. (%) [A] | Evaluation of IND. [B] | Value (%) [A]×[B] |
|---|---|---|---|---|
| C4. Materials, Solid Residues/Resources management | Reuse/Recycle Materials | 1.50% | 0—E | 0% |
| | Environmental Management Plan | 2.00% | 1.1—A* | 2.20% |
| | Flexibility/Adaptability | 1.50% | 0.34—C | 0.50% |
| Sum | | 5.00% | | 2.70% |

Based on this calculation, the value given to this category is 0.54%.

C5. Category: Water

I11. Water Consumption

This indicator is calculated through a form that evaluates the average water consumption of each interior and exterior device, the annual consumption of water for drinking and irrigation [15,16,18]. When the school was refurbished, Empresa Parque Escolar was very concerned about the use of toilets, taps and showers with the aim of reducing consumption [21].

$$\overline{P_{WC}} = \frac{P_{WC} - P_{WC*}}{P_{WC}^* - P_{WC*}}, \overline{P_{WC}} = \frac{9,919,000 - 19,551,550}{8,780,087 - 19,551,550} = 0.89\% \tag{14}$$

$\overline{P_{WC}}$—Normalized value of the Indicator Water Consumption.

Based on this calculation, the value given to this indicator is 0.89%.

I12. Water Treatment and Recycling

This indicator is related to the annual per capita (l/year) use of procedures related to the recycling system [15,16,18]. Through the analysis of the materials elaborated by the Empresa Parque Escolar (EPE), it was observed that there is no specific concern about this subject [21]. Therefore, there are no concerns related to the treatment or recycling related to water. Based on this information, the value given to this indicator is 0%.

I13. Collection and Reuse of Rainwater

This indicator is related to the value of the capacity of the building to manage the reuse of rainwater (l/year). [15,16,18]. Through the analysis of the materials elaborated by the Empresa Parque Escolar (EPE), it was concluded that there is no concern about this subject [21]. Therefore, there is no treatment related to rainwater management. Based on this information, the value given to this indicator is 0%.

Calculation of Category 5

In Table 6, the percentage of Category 5 is calculated, by the sum of the indicators included in this category, and the percentage relative to the total value, 1.

$$\sum \text{Value } [A] \times [B] = \sum \text{Indicator Evaluation } [A] \text{of C5, C5 1, } 2.67 = 7.00 \text{ C5} = 0.38\%, \text{ C5 1} \quad (15)$$

**Table 6.** Calculation of the percentage of Category 5.

| Category | Indicator | Weight of IND. (%) [A] | Evaluation of IND. [B] | Value (%) [A]×[B] |
|---|---|---|---|---|
| | Water Consumption | 3.00% | 0.89—A | 2.67% |
| C5. Water | Water Treatment and Recycling | 3.00% | 0—E | 0% |
| | Collection and Reuse of Rainwater | 1.00% | 0—E | 0% |
| Sum | | 7.00% | | 2.67% |

Based on this calculation, the value given to this category is 0.38%.

Calculation of the Environmental Dimension

In Table 7, the percentage concerning the Environmental Dimension is calculated, by the sum of the categories included in this dimension, and the percentage relative to the total value, 1.

$$\sum \text{Value } [A] \times [B] = \sum \text{Indicator Evaluation } [A] \text{ of ED, ED 1, } 21.98 = 35.00 \text{ ED} = 0.63\%, \text{ ED 1} \quad (16)$$

**Table 7.** Calculation of the percentage of Environmental Dimension.

| Dimension | Category | Weight of CAT. (%) [A] | Evaluation of CAT. [B] | Value (%) [A]×[B] |
|---|---|---|---|---|
| | C1. Climate Change/Air Quality | 7.00% | 0.76—B | 5.32% |
| | C2. Biodiversity and Land Use | 5.00% | 0.50—B | 2.50% |
| Environment AL (40%) | C3. Energy | 11.00% | 0.80—A | 8.80% |
| | C4. Materials, Solid Residues/Resources Management | 5.00.% | 0.54—C | 2.70% |
| | C5. Water | 7.00% | 0.38—C | 2.66% |
| Sum | | 35.00% | | 21.98% |

Based on this calculation, the value given to the Environmental Dimension is 0.63%

Social Dimension

C6. Category: User Health and Comfort

I14. Indoor Air Quality

This indicator is related to the renewal rate of the air in the building and to the finishing materials with Volatile Organic Compounds (VOC) content, having also been calculated through a form that evaluates all the indicators of comfort.

One of the objectives of the EPE is "to improve living conditions and environmental comfort, with particular emphasis on hygrothermics, acoustics, and air quality" [21]. Much of the information necessary for the analysis of air quality required by this methodology was not found, such as the air renewal rate expected for the construction, and the finishing materials with VOC content, according to the standards established by the RCE [28]. Based on this information, the value given to this indicator is 1.00%.

I15. Thermal Comfort

This indicator has to do with the level of thermal comfort during the winter and summer seasons and was also calculated through a form that evaluated all the comfort indicators [15,16,18]. The temperature of all the environments of this school is controlled by cooling or heating system;

always maintaining the ideal temperature. Based on this information, the value given to this indicator is 1.00%.

I16. Visual Comfort

This indicator is related to the illuminance levels provided by natural or artificial lighting in every compartments of the building where there is occupation. It also uses a form that evaluates all the comfort indicators [15,16,18]. The classrooms are mostly in the North and South facades, receiving natural lighting for practically the whole day. Laboratories, support rooms, circulation and others, which are less used, receive a reduced amount of sunlight.

$$\overline{E} = \frac{E - E_*}{E^* - E_*}, \frac{88\% - 10\%}{20\% - 10\%} = 0.78\% \tag{17}$$

$\overline{E}$—Normalized value of the Indicator Visual Comfort.

Based on this information, the value given to this indicator is 0.78%.

I17. Acoustic Comfort

This indicator refers to the reverberation time in the classroom, the level of acoustic over percussion sounds and the comfort to airborne sounds concerning classrooms [15,16,18]. It also uses a form that evaluates all the comfort indicators. In the context of the renovation of the school, several procedures were done with the intention of solving existing acoustic problems and avoiding new problems, mainly related to the acoustic isolation between the external and the internal environments.

$$\overline{D_{2m,nT,w}} = \frac{D_{2m,nT,w} - D_{2m,nT,w*}}{D^*_{2m,nT,w} - D_{2m,nT,w*}}, \frac{34.4\% - 30\%}{36\% - 30\%} = 0.73\% \tag{18}$$

$\overline{D_{2m,nT,w}}$—Normalized value of the Indicator Acoustic Comfort.

Based on this calculation, the value given to this indicator is 0.73%.

I18. Ergonomic Comfort

This indicator deals with the comfort of the school desks. It uses a form that evaluates all the comfort indicators related to ergonomics, specifically about the proper sizing of the desks where students sit. Through the analysis of the materials elaborated by the Empresa Parque Escolar (EPE), it can be observed that there is no concern about this subject [21]. It is not possible to define specific physical characteristics for students between the ages of 14 and 18, for a very large variety of sizes. Therefore, it is not possible to define a standard dimension for school desks. A questionnaire was applied to students in the Francisco de Holanda high school about the level of discomfort related to ergonomic comfort, specifically about the proper sizing of the chair and table. The result of these questionnaires is reported in [22]. The result of this indicator is based on the result of this research, shown below.

$$E = \frac{\sum \textit{Credits obtained}}{\sum \textit{Total credits}}(\%), \frac{60}{100}(\%) = 60\% \tag{19}$$

$\overline{E}$—Normalized value of the Indicator Ergonomic Comfort.

$$\overline{E} = \frac{E - E_*}{E^* - E_*}, \frac{60\% - 10\%}{20\% - 10\%} = 0.50\% \tag{20}$$

Based on this calculation, the value given to this indicator is 0.50%.

Calculation of Category 6

In Table 8, the percentage of Category 6 is calculated, by the sum of the indicators included in this category, and the percentage relative to the total value, 1.

$$\sum \text{Value } [A] \times [B] = \sum \text{Indicator Evaluation } [A] \text{ of C6}, \text{ C6 1, 21.08} = 25.00 \text{ C6} = 0.84\%, \text{ C6 1} \tag{21}$$

**Table 8.** Calculation of the percentage of Category 6.

| Category | Indicator | Weight of IND. (%) [A] | Evaluation of IND. [B] | Value (%) [A]×[B] |
|---|---|---|---|---|
| C6. User Health and Comfort | Indoor Air Quality | 6.00% | 1.0—B | 6.00% |
| | Thermal Comfort | 5.50% | 1.0—A | 5.50% |
| | Visual Comfort | 6.00% | 0.78—A | 4.68% |
| | Acoustic Comfort | 5.00% | 0.73—A | 3.65% |
| | Ergonomic Comfort | 2.50% | 0.50—B | 1.25% |
| Sum | | 25.00% | | 21.08% |

Based on this calculation, the value given to this category is 0.84%.

C7. Category: Accessibility

I19. Mobility Plan

This indicator is calculated through a form that assesses the conditions of access to the school on foot, by bicycle, as well as access for disabled persons [15,16,18].

Through the analysis of the materials elaborated by the Empresa Parque Escolar (EPE), it was concluded that there is no concern about this subject [21]. In this school, there is no area dedicated to cyclists, such as parking or routes, which makes the value of this indicator very low. However, there is a wide variety of public transport near the school. Several compartments of the building are designed to aid the access of people with disabilities.

$$P_M = \frac{\sum \text{ Credits obtained}}{\sum \text{ Total credits}}(\%), \frac{40}{100}(\%) = 40\% \tag{22}$$

$\overline{P_M}$—Normalized value of the Indicator Mobility plan.

$$\overline{P_M} = \frac{P_M - P_{M*}}{P_M^* - P_{M*}}, \frac{40\% - 10\%}{90\% - 10\%} = 0.37\% \tag{23}$$

Based on this calculation, the value given to this indicator is 0.37%.

Calculation of Category 7

In Table 9, the percentage of Category 7 is calculated, by the sum of the indicators included in this category, and the percentage relative to the total value, 1.

$$\sum \text{Value } [A] \times [B] = \sum \text{Indicator Evaluation } [A] \text{ of C7, C7 1, } 0.74 = 2.00 \text{ C7} = 0.37\%, \text{ C7 1} \tag{24}$$

**Table 9.** Calculation of the percentage of Category 7.

| Category | Indicator | Weight of IND. (%) [A] | Evaluation of IND. [B] | Value (%) [A]×[B] |
|---|---|---|---|---|
| C7. Accessibility | I22 Mobility Plan | 2.00% | 0.37—C | 0.74% |

Based on this calculation, the value given to this Category is 0.37%.

C8. Category: Security and Safety

I20. Occupants Security and Safety

This indicator is calculated through a form that assesses the guarantee of the proper functioning of the main services of the building, such as water, energy, telecommunications [15,16,18] and the protection of students from being harmed. One of the objectives of the Empresa Parque Escolar is "to improve, security and accessibility" [21]. Several measures were taken regarding the safety of students in school buildings.

$$\overline{P_{SO}} = \frac{P_{SO} - P_{SO*}}{P_{SO}^* - P_{SO*}}, \frac{100\% - 30\%}{90\% - 30\%} = 1.20\% \tag{25}$$

$\overline{P_{SO}}$—Normalized value of the Indicator Occupants Security and Safety.

Based on this calculation, the value given to this indicator is 1.20%.

Calculation of Category 8

In Table 10, the percentage of Category 8 is calculated, by the sum of the indicators included in this category, and the percentage relative to the total value, 1.

$$\sum \text{Value } [A] \times [B] = \sum \text{Indicator Evaluation } [A] \text{ of C8, C8 1, } 3.60 = 3.00C8 = 1.20\%, \text{ C8 1} \quad (26)$$

**Table 10.** Calculation of the percentage of Category 8.

| Category | Indicator | Weight of IND. (%) [A] | Evaluation of IND. [B] | Value (%) [A]×[B] |
|---|---|---|---|---|
| C8. Security and Safety | I20 Occupants Security and Safety | 3.00% | 1.20—A* | 3.60% |

Based on this calculation, the value given to this category is 1.20%.

C9. Category: Education for Sustainability Awareness

I21. Sustainability Awareness

This indicator is obtained through a form that evaluates the level of the students regarding sustainability awareness. Through the analysis of the materials elaborated by the Empresa Parque Escolar (EPE), it is concluded that there is no concern about this subject [21]. Based on this calculation, the value given to this indicator is 0.57.

$$\overline{L_{SA}} = \frac{L_{SA} - L_{SA*}}{L_{SA}^* - L_{SO*}}, \frac{60\% - 10\%}{20\% - 10\%} = 0.78\% \quad (27)$$

$\overline{L_{SA}}$—Normalized value of the Indicator Sustainability Awareness.

Based on this calculation, the value given to this indicator is 0.78%.

Calculation of Category 9

In Table 11, the percentage of Category 9 is calculated, by the sum of the indicators included in this category, and the percentage relative to the total value.

$$\sum \text{Value } [A] \times [B] = \sum \text{Indicator Evaluation } [A] \text{ of C9, C9 1, } 1.71 = 3.00C9 = 0.57\%, \text{ C9 1} \quad (28)$$

**Table 11.** Calculation of the percentage of Category 9.

| Category | Indicator | Weight of IND. (%) [A] | Evaluation of IND. [B] | Value (%) [A]×[B] |
|---|---|---|---|---|
| C9 Education for Sustainability Awareness | I21 Sustainability Awareness | 3.00% | 0.57—B | 1.71% |

Based on this calculation, the value given to this category is 0.57%.

C10. Category: Sustainability of the Area

I22. Accessibility to Public Transport

This indicator is calculated through a form that evaluates the type of urban area, the waiting time, frequency, travel time and total access time for each public transport line and the distance between the main entrance and each public transport stop [15,16,18].

Through the analysis of the materials elaborated by the Empresa Parque Escolar (EPE), it was detected that there is no concern about this subject [21]. There is a bus stop in front of the school, where a great diversity of lines passes. The train station is 1 km from the school.

$$I_{ATP} = 2.2 + 1.9 + 2.0 = 6.1 \qquad (29)$$

$\overline{I_{ATP}}$—Normalized value of the Indicator Sustainability of the area.

Normalized value of accessibility to public transport.

$$\overline{I_{ATP}} = \frac{I_{ATP} - I_{ATP*}}{I^*_{ATP} - I_{ATP*}}, \frac{6.1\% - 2\%}{7.5\% - 2\%} = 0.75\% \qquad (30)$$

Based on this calculation, the value given to this indicator is 0.75%.

Calculation of Category 10

In Table 12, the percentage of Category 10 is calculated, by the sum of the indicators included in this category, and the percentage relative to the total value.

$$\sum \text{Value } [A] \times \ [B] = \sum \text{Indicator Evaluation } [A] \text{ of } C10, C10 \ 1, \ 1.50 = 2.00 C10 \\ = 0.75\%, C10 \ 1 \qquad (31)$$

**Table 12.** Calculation of the percentage of Category 10.

| Category | Indicator | Weight of IND. (%) [A] | Evaluation of IND. [B] | Value (%) [A]×[B] |
|---|---|---|---|---|
| C10 Sustainability of the Area | I23. Accessibility to Public Transport | 2.00% | 0.75—A | 1.50% |

Based on this calculation, the value given to this category is 0.75%.

Calculation of the Social Dimension

In Table 13, the percentage of Social Dimension is calculated, by the sum of the categories included in this dimension, and the percentage relative to the total value, 1.

$$\sum \text{Value } [A] \times \ [B] \ = \sum \text{Indicator Evaluation } [A] \text{ of } SD, SD \ 1, \ 28.63 = 35.00 SD \\ = 0.82\%, SD \ 1 \qquad (32)$$

**Table 13.** Calculation of the percentage of Social Dimension.

| Dimension | Category | Weight of IND. (%) [A] | Evaluation of IND. (%) [B] | VALUE (%) [A]×[B] |
|---|---|---|---|---|
| | C6. User Health and Comfort | 25.00% | 0.84—A | 21.08% |
| | C7. Accessibility | 2.00% | 0.37—C | 0.74% |
| SOCIETY 35% | C8. Security and Safety | 3.00% | 1.20—A* | 3.60% |
| | C9. Education for Sustainability Awareness | 3.00% | 0.57—B | 1.71% |
| | C10: Sustainability of the Area | 2.00% | 0.75—A | 1.50% |
| Sum | | 35.00% | | 28.63 |

Based on this calculation, the value given to Social Dimension is 0.82%.

Economic Dimension

C11. Category: Life Cycle Costs

I23. Life Cycle Costs

Through the analysis of the materials elaborated by the Empresa Parque Escolar (EPE), it was observed that there is no concern about this subject [21]. However, every company, state-owned or private, has concerns about the financial part. Only social or sustainable concerns are not sustainable if the building does not sustain itself economically. This indicator is related to the performance of the

building related to initial cost and operating costs (water consumption and energy). For the evaluation of this indicator, the values of purchase and sale of the property are analyzed [15,16,18].

With regard to the case of the Francisco de Holanda high school, it was not possible to assess the value of the sale because this is a public school. The value of the purchase was also not possible to be evaluated, since the land was bought in the nineteen century and the school began to be constructed in 1886, having new reforms in 1959 and in 2011 [30].

In the case of this school, the amount paid by the government relative to the number of students, together with the monthly capital spent with aspects such as energy, gas and water bills, among others, was analyzed. These spending must be made in a way that guarantees the health and well-being of the students in terms of thermal comfort, light, air quality, water consumption and others.

According to the research made by Saraiva et al. about the Francisco de Holanda high school [22], most of the students (88%) are comfortable or a little uncomfortable with regard to the environmental comfort. The studies related to the categories 3 and 5 of this work demonstrate that water and energy consumption are adequate. After verifying this data, it was noticed that the monthly budget used for all expenses of the Francisco de Holanda high school does not exceed the budget, and this meets the basic requirements for student comfort. It was not possible to perform a mathematical calculation as indicated by the methodology of this indicator. Due to this impossibility and the information described above, the value given to this indicator is 1.

Calculation of Category 11

In Table 14, the percentage of Category 11 is calculated, by the sum of the indicators included in this category, and the percentage relative to the total value, 1.

$$\sum \text{Value} [A] \times \ [B] = \sum \text{Indicator Evaluation} [A] \text{ of C11}, \text{C11 1}, \ 30.00 = 30.00 \text{C11} \\ = 1.00\%, \ \text{C11 1} \tag{33}$$

**Table 14.** Calculation of the percentage of Category 11.

| Category | Indicator | Weight of IND. (%) [A] | Evaluation of IND. [B] | Value (%) [A]×[B] |
|---|---|---|---|---|
| C11 Life Cycle Costs | I26. Life Cycle Costs | 30.00% | 1.00—A* | 30.00% |

Based on this calculation, the value given to this category is 1.00%.

Calculation of the Economic Dimension

In Table 15, it is calculated the percentage of the Economic Dimension, by the sum of the categories included in this dimension, and the percentage relative to the total value, 1.

$$\sum \text{Value} [A] \times \ [B] = \sum \text{Indicator Evaluation} [A] \text{ of ED}, \text{ED 1}, \ 30.00 = 30.00 \text{ES} \\ = 1.00\%, \ EcD = 1, \ \text{Economic Dimension} = 1.00\% \tag{34}$$

**Table 15.** Calculation of the percentage of the Economic Dimension.

| Dimension | Category | Weight of IND. (%) [A] | Evaluation of IND. [B] | Value (%) [A]×[B] |
|---|---|---|---|---|
| ECONOMIC 30% | C11 Life Cycle Costs | 30.00% | 1.00—A | 30.00% |
| Sum | | 30.00% | | 30.00% |

In summary, Table 16 presents the results obtained in the evaluation of all sustainability indicators of SAHSB$^{PT}$ in the Francisco da Holanda High School.

**Table 16.** Summary of the analysis and demonstration of the results.

| Dimension | Category | Indicator | Weight of IND. (%) [A] | Evaluation of IND. [B] | Value (%) [A]×[B] |
|---|---|---|---|---|---|
| ENVIRONMENTAL (good practice = 35.00%) | C1. Climate Change and Air Quality | I1. Life Cycle Environmental Impacts | 4.00% | 0.75—C | 3.00% |
| | | I2. Heat Island Effects | 3.00% | 0.77—A | 2.31% |
| | C2. Biodiversity and Land Use | I3. Land Use Efficiency | 4.00% | 0.63—B | 2.52% |
| | | I4. Product With Organic Certificate | 1.00% | 0—E | 0% |
| | C3. Energy | I5. Energy Consumption | 5.00% | 0.8—A | 4.00% |
| | | I6. Renewable Energy | 3.00% | 0.6—B | 1.80% |
| | | I7. Commissioning | 3.00% | 1.00—A* | 3.00% |
| | C4. Materials, Solid Residue/Resource Management | I8. Reuse and Recycle of Materials. | 1.50% | 0—E | 0% |
| | | I9. Environmental Management Plan | 2.00% | 1.10—A* | 2.20% |
| | | I10. Flexibility and Adaptability | 1.50% | 0.34—C | 0.51% |
| | C5. Water | I11. Water Consumption | 3.00% | 0.89—A | 2.67% |
| | | I12. Water Treatment and Recycling | 3.00% | 0—E | 0% |
| | | I13. Collection and Reuse of Rainwater | 1.00% | 0—E | 0% |
| | | Sum or total | | | 22.01% |
| SOCIAL (good practice =35.00%) | C6. User Health and Comfort | I14. Indoor Air Quality | 6.00% | 1.00—B | 6.00% |
| | | I15. Thermal Comfort | 5.50% | 1.00—A | 5.50% |
| | | I16. Visual Comfort | 6.00% | 0.78—A | 4.68% |
| | | I17. Acoustic Comfort | 5.00% | 0.73—A | 3.65% |
| | | I18. Ergonomic Comfort | 2.50% | 0.50—B | 1.25% |
| | C7. Accessibility | I19. Mobility Plan | 2.00% | 0.37—C | 0.74% |
| | C8. Security and Safety | I20. Occupants Security and Safety | 3.00% | 1.20—A* | 3.60% |
| | C9. Education for Sustainability Awareness | I21. Sustainability Awareness | 3.00% | 0.57—B | 1.71% |
| | C10. Sustainability of the Area | 122. Accessibility to Public Transport | 2.00% | 0.75—A | 1.50% |
| | | Sum or total | | | 28.63% |
| ECONOMY (good practice = 30.00%) | C11. Life Cycle Costs | I23. Life Cycle Costs | 30.00% | 1.00—A | 30.00% |
| | | Sum or total | | | 30.00% |

The analysis of Table 16 shows that most of the indicators, 35%, achieved a level A and 24% achieved a level B. Just 15% of the indicators achieved a level E and 13% achieved a level A+ and C. These results demonstrate that more than 72% of the indicators reached a level A+, A and B. Therefore, this is a good result.

The analysis of Table 16 shows that most of the categories (82%) were evaluated with A and with B, 18% with E, A+ and C. These results demonstrated that more than 80% of the categories are A and B, therefore, this is a good result.

The overall Sustainability Level (SL) of the building and the weight considered for each environmental dimension is shown in Table 17.

**Table 17.** Dimensions and sustainability levels.

| Dimension | Quantitative | Qualitative |
|---|---|---|
| Environmental | 22.05% | 0.63—B |
| Social | 28.70% | 0.82—A |
| Economic | 30.00% | 1.00—A |
| Sustainability level | 80.75% | 0.81—A |

The analysis of Table. 17 shows that most of the dimension (66%) were evaluated with A and 34% with B, therefore, it is an outstanding result. The total value is 0.81, which corresponds to the qualitative level of sustainability "A".

## 4. Conclusions

Empresa Parque Escolar has recently renovated the high school building studied in this thesis, specifically from 2011 to 2013. The Portuguese government intends to reform most of the Portuguese schools with the support of Empresa Parque Escolar, with a single standard. The EPE intervened in 477 high schools of the 616 existing Portuguese high schools [21] which represents 77.5%. Considering that Francisco de Holanda High School (the chosen school) has the standards determined by EPE, it therefore represents Portuguese schools.

Some of the indicators in this methodology had already been part of the standard rules of Empresa Parque Escolar, however, others are new subjects. The average performance of the Francisco de Holanda high school is 75%. Most of the indicators, 82%, got an A or B grade. Since this is one of the high schools built by Empresa Parque Escolar, and that the company intends to reform 74% of the high schools in Portugal under the same rules, this shows that Portuguese high schools tend to have a good result.

It must be noted that, although the standards of construction and reform are established by the EPE, to know the result achieved by each particular school, the global methodology [14,22] must be applied to each school, which includes the application of the questionnaires to the students, the measurement of key indicators and the consultation of the school management reports, taking also into account the climate context, the type of construction as well as the urban context. These results demonstrated that more than 82% of the categories achieved a level of at least category B, demonstrating that the Empresa Parque Escolar is achieving its objectives related to the habitability and environmental comfort conditions (air and thermic quality) and accessibility, to modernize equipment and safety, besides guaranteeing energy efficiency and the flexibility and adaptablity of the school buildings. The results for Social (0.80—Best Practice) and Economic (1—Best Practice) dimensions are A, whereas for the Environmental Dimension (0.63) the grade is B.

The results of the methodology applied in this school are good in most of the indicators and categories. The results of the indicators demonstrated that more than 74% are A+, A and B.

The school evaluated represents the Portuguese standard high school, thus, this represents a good result regarding Portuguese high schools, demonstrating how high schools are improving in terms of sustainability.

**Supplementary Materials:** The following are available online at http://www.mdpi.com/2071-1050/11/17/4559/s1.

**Author Contributions:** T.S. undertook the main body of research to elaborate this article, and also developed the investigation method and analyzed the results with the support of M.A. This article was written with the contributions from M.A., L.B., and M.T.B. M.A. and M.T.B. aided to shape the discussion sections of the article and provided critical judgment on this research. All authors read and approved the final manuscript.

**Funding:** This research received no external funding.

**Acknowledgments:** The corresponding author wishes to thank Empresa Parque Escolar (EPE) for supplying the necessary material for the execution of the work; the teacher and Administrative Director of the Francisco de Holanda High School, Abílio Ferreira, for showing me the school and for giving all the necessary

information. This research did not receive any specific grant from funding agencies in the public, commercial, or not-for-profit sectors.

**Conflicts of Interest:** The authors declare that there is no conflict of interest.

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
