# Peer review of "Verification of the Adequacy of the Portuguese Sustainability Assessment Tool of High School Buildings, SAHSBPT, to the Francisco de Holanda High School, Guimarães"

_sustainability, doi:10.3390/su11174559_

Round 1

Reviewer 1 Report

This article shows the application of the SBTool methodology developed specifically for Portuguese high schools: SAHSBpt. This methodology has been applied to the Francisco de Holanda High School Building, in Guimarães, Portugal.

There are many drawbacks in the conversion from word to pdf that make a very difficult reading.

The application of the chosen methodology generates an excessive repetition of the procedure. I suggest presenting the formula of Diaz-Baltero only once at the beginning,  expressing the indicators and providing the results, directly.

Moreover, the evaluation of the economic dimension seems to me too approximate.

Many repetitions in the text. Read it again carefully. And beyond:

line 67: insert bracket

line 68: number is better, 23%

line 70: the same, 46%

line 192: it has been said before in line 87

line 233 nd many others: sometimes you use coma, sometimes point. Do a choice, please

line from 247 to 250: some misunderstanding

Some mistakes in the references

n. 4. the title is wrong. The correct one is "environmental responsibility in building design: an italian regional study"

n. 5 some mistakes in the format

Reviewer 2 Report

The research meets with the aim and subject of the Journal.  It is a timely topic.

It involves the application of a sustainability assessment tool to a case study. The tool is defined in another article published in a magazine of MDPI publishing, Applied Science, on June 30, 2019, 1 month before the review of this article.

This article states that the tool was tested at the Francisco de Holanda school to verify the adequacy of the comparative analysis and its applicability to the context of schools in Portugal. It is in this article this reviewer is analyzing that the case study assessment is described.

The authors must indicate in this article that the evaluation tool is their own and that it is defined in the aforementioned article.

In the last paragraph of the introduction section of the article under review it is stated that the objective is to verify if the indicators of the tool are adequate, and to know the situation of the schools in Portugal, being the Francisco de Holanda representative of this type of buildings.

TITLE: Application of the Portuguese Sustainability Assessment Tool of High School buildings, SAHSBPT, to the Francisco de Holanda High School, Guimarães.

The title is clear, concise, specific and relevant, and it synthesizes the object and objectives of the investigation. However, the objective of a research should not be the application of a tool, but the verification or assessment of the adequacy/efficiency/ applicability of it. Please consider modifying the title.

ABSTRACT:

Authors should review the use of semicolon. Abstract should include the article's main findings and conclusions.

KEYWORDS:

sustainability assessment tools; SBTool; high school buildings; Portugal;

In the opinion of the reviewer the name of the reference tool should not be included in the list of Keywords. Please, avoid acronym/abbreviations in Abstract and Keywords.

INTRODUCTION

Please, the description of the case study should be in a section 2. Materials and Methods, not in the introduction.

STRUCTURE OF THE SAHSB METHODOLOGY

This section should be 2. Materials and methods, and the description of the tool a subsection. In addition, the structure of the methodology published in a previous article should be specifically referenced.

Please, describe better the school, as well as the context, including regulatory, climate, social, and economic context, constructive issues, facades orientation. Have the authors carried on the research with the participation of the community? Did they take data on site?

Please describe the methodology not only the tool.

RESULTS AND DISCUSSION

The description of the case study should be improved, including:

weather data constructive systems urban context dates of the different constructions constructed throughout history facade orientation Public or private center building use schedule

In Section 3.2 the authors indicate that Diaz Balteiro's formula was used. Please describe this formula and its philosophy so to understand the process.

Indicators I14, I15, I16 and I17 are very important issues in the assessment of a case study and should be measured on site (not so if it were the evaluation of a project (not built)). Authors should describe the construction systems of facades, partitions, dimensions of windows, if they have sun protection, type of interior surfaces of the classrooms, etc.

The ergonomic comfort is also an important issue, but if it is not possible to assess, value should be removed, not considered.

The reviewer of this article has tried to understand these questions by reading the article of the journal applied science but has not found in it a description that allows to evaluate the evaluation procedure performed.

The analysis of a case study of a building in use must be supported by real measurements in situ, not just with formulas. Only this way the efficiency of the tool could be checked.

CONCLUSIONS

In the last paragraph of the introduction section of the article under review it is stated that the objective is to verify if the indicators of the tool are adequate, and to know the situation of the schools in Portugal, being the Francisco de Holanda representative of this type of buildings.

In the opinion of the reviewer it is not possible to conclude that the case study is representative of the Portuguese schools, as it depends on the climate context, the construction type, and the urban context.

The authors could only verify the applicability of the theoretical evaluation and detect aspects not considered or poorly evaluated, but to verify the suitability of the tool it can only be done by comparing it with real measurements.

As most of the issues studied are theoretical and not real, it is not possible to state that the building responds to the requirements of comfort, health and safety.

Reviewer 3 Report

The title is interesting and significant, and the research has been performed appropriately. The paper is very good from a scientific point of view and the methodology used. The output is current, the conclusions are correct. The English and the arrangement of the paper is sound, and its flow makes it readable and comprehensible.

Author Response

The authors would like to thank the reviewer for the comments

Round 2

Reviewer 1 Report

The paper can be accepted in this form

Reviewer 2 Report

Thank you very much for considering the comments.

I will accept in present form.